# Biochemical Screening for Fetal Trisomy 21: Pathophysiology of Maternal Serum Markers and Involvement of the Placenta

**DOI:** 10.3390/ijms24087669

**Published:** 2023-04-21

**Authors:** Jean Guibourdenche, Marie-Clémence Leguy, Guillaume Pidoux, Marylise Hebert-Schuster, Christelle Laguillier, Olivia Anselem, Gilles Grangé, Fidéline Bonnet, Vassilis Tsatsaris

**Affiliations:** 1Hormonologie CHU Cochin AP-HP, 75014 Paris, France; 2Faculté de Santé, Université Paris Cité, 75014 Paris, France; 3FHU Préma, 75014 Paris, France; 4INSERM UMR-S1180, 75014 Paris, France; 5UMR-S1139, 75014 Paris, France; 6Maternité Port Royal CHU Cochin AP-HP, 75014 Paris, France

**Keywords:** hCG, hCG free β subunit, inhibin A, PAPP-A, unconjugated estriol, placenta, cell free fetal DNA, prenatal screening, fetal aneuploidy, maternal blood

## Abstract

It is now well established that maternal serum markers are often abnormal in fetal trisomy 21. Their determination is recommended for prenatal screening and pregnancy follow-up. However, mechanisms leading to abnormal maternal serum levels of such markers are still debated. Our objective was to help clinicians and scientists unravel the pathophysiology of these markers via a review of the main studies published in this field, both in vivo and in vitro, focusing on the six most widely used markers (hCG, its free subunit hCGβ, PAPP-A, AFP, uE3, and inhibin A) as well as cell-free feto–placental DNA. Analysis of the literature shows that mechanisms underlying each marker’s regulation are multiple and not necessarily directly linked with the supernumerary chromosome 21. The crucial involvement of the placenta is also highlighted, which could be defective in one or several of its functions (turnover and apoptosis, endocrine production, and feto–maternal exchanges and transfer). These defects were neither constant nor specific for trisomy 21, and might be more or less pronounced, reflecting a high variability in placental immaturity and alteration. This explains why maternal serum markers can lack both specificity and sensitivity, and are thus restricted to screening.

## 1. Introduction

Trisomy 21, also called Down syndrome, is a chromosomal abnormality defined by the presence of a supernumerary chromosome 21 either completely or partially [1]. In 94% of cases, it is a free and homogeneous trisomy 21 affecting all cells and resulting from a meiotic nondisjunction. The clinical syndrome was described for the first time by Esquirol in 1833 and Down in 1886, and the association of this syndrome with the presence of an additional chromosome 21 was established by Gautier and Lejeune in 1956 [2]. Trisomy 21 phenotype is variable [3]. It associates constant mental retardation with anatomical malformations, some of them being detectable with fetal ultrasound scanning (e.g., heart disease, duodenal stenosis or atresia, etc.). Indication for prenatal screening of fetal trisomy 21 was initially only based on maternal age, as the risk of aneuploidy increases with age [4]. Since then, fetal ultrasound scanning for those fetal abnormalities during the second trimester of gestation and, presently, increased nuchal translucency (NT) during the first trimester of gestation has become a major component of prenatal screening [5,6].

Maternal serum markers (MSM) during the second trimester, i.e., human chorionic gonadotropin (hCG) and its free β subunit, alpha feto protein (AFP), unconjugated estriol (uE3), and inhibin A, were described as abnormal in trisomy 21 in the 1990s [7,8,9,10,11]. Indeed, hCG and inhA are increased by about two multiples of the median value (MoM), and AFP and uE3 are decreased by about 0.8 MoM in trisomy 21-affected pregnancies. First-trimester markers, i.e., a free β subunit of hCG (free hCGβ) and pregnancy-associated plasmatic protein A (PAPP-A) have also been described as abnormal [12,13]. The mean pattern in trisomy 21-affected pregnancies associates an increased NT by 2.5 MoM, an increased free hCGβ by 2.2 MoM, and a decreased PAPP-A by 0.5 MoM with a mean detection rate of 84% with a 5% false positive rate [14,15]. Over recent years, noninvasive prenatal testing (NIPT) has spread widely. Indeed, the detection of cell-free feto–placental DNA in maternal blood is both far more sensitive and specific than MSM, with a detection rate of >99% and a false positive rate of <1% [16].

However, little is known about the mechanisms underlying these abnormal maternal serum profiles. Accordingly, we aim to unravel the physiology of these markers (PAPP-A, free hCGβ, hCG, AFP, uE3, inhA, and cell-free feto–placental DNA) and their regulation, focusing on the placenta and try to explain their abnormal variations in maternal serum in trisomy 21-affected pregnancies.

## 2. Soluble Markers

### 2.1. Markers of Placental Origin

From the beginning of pregnancy, as soon as the blastocyst, which will further differentiate into embryo and placenta, makes contact with the maternal uterine circulation, it starts producing markers that are soon detectable in the maternal bloodstream [17,18]. Therefore, many of these maternal serum markers, such as hCG, free hCGβ, inhibin A, and PAPP-A, are strictly of placental origin, as they are produced by the trophoblast, the specific placental tissue [19].

#### 2.1.1. Human Chorionic Gonadotropin

The human chorionic gonadotropin (hCG) hormone was discovered in 1920 [20]. It is a dimeric glycoprotein (MW 40–45 kDa) that belongs to the family of gonadotropins such as LH and FSH, or stimuli such as TSH produced by the pituitary gland [21]. It results from the non-covalent binding of a common α subunit and a specific ß subunit, forming a dimeric glycoprotein that acts on a receptor, the LH/CG-R [22,23] (Figure 1A). The α subunit (hCGα) is encoded by a single gene located on chromosome 6, comprising 92 amino acids and five disulfide bonds (MW 14.5 kDa), and two N glycan chains [24,25]. The β subunit (hCGβ) is specific and confers its biological specificity to each hormone. It is encoded by a cluster of genes located on chromosome 19 and subdivided into type I (ß6/7 CG), and type II (ß3/9 CG, ß5 CG, and ß8 CG) genes, the latest being predominantly expressed in the placenta [26,27]. Among gonadotropins, hCGβ is the longest β subunit (145 amino acids, MW 22 kDa) with six disulfide bonds, and the most glycosylated with two N glycans and four O glycans [25,28].

Glycosylation impacts hCG clearance and biological activity, and is likely to depend on the origin of the producing cells, i.e., pituitary cells, normal placental, and placental tumors [29,30,31,32]. The extravillous cytotrophoblast (EVCT) produces high levels of hyperglycosylated hCG that are likely to modulate trophoblastic invasion and angiogenesis while the syncytiotrophoblast (ST) is the principal placental source of hCG [33,34,35,36,37,38]. 

hCG production in vitro is regulated by factors that control trophoblast differentiation via cAMP: hormones (e.g., corticoids, progesterone, estradiol.), hCG itself, neuropeptides (e.g., CRH), growth factors and cytokines (e.g., EGF, TNFα), PPARγ, and oxygen [37,39]. In vivo, the secretory peak of hCG during the first trimester remains unexplained. Several hypotheses have been proposed: the stimulation of hCG by GnRH or the dominant presence of a truncated LHCGR in early pregnancy [40,41], and thereafter apoptosis of hCG-secreting trophoblast cells [42] or negative feedback on hCG synthesis [43]. Once produced, hCG may be truncated under the action of placental proteases, leading to unstable “nicked hCG”, which rapidly dissociates into hCGα and truncated hCGβ (hCGβn) [44]. During gestation, free hCGα levels rise in the maternal circulation in line with placental mass, while hCG and free hCGβ follow the same profile, peaking at around 10-12 WG (Figure 1A). A total of 80% of serum circulating hCG is metabolized by the maternal liver, whereas 20% is degraded and excreted by the kidneys [45]. The main final degradation product is the β core (hCG-βcf), a small 9 kDa fragment of hCG [46]. The clearance rate of hCG in serum is best described using a three-component exponential function, leading to a global half-life for hCG of between 24 and 36 h. The two free subunits have shorter half-lives of 2 h for the free α subunit and 4 h for the free β subunit [47]. 

hCG and its glycoforms are critical hormones during pregnancy, as they are involved in placental implantation and invasion, and regulate different fetal and maternal–placental functions [29,30,48,49]. hCG takes over from maternal pituitary LH during the first trimester of gestation, allowing the maintenance of the ovarian corpus luteum and its transformation into a gravid corpus luteum, producing 17αOH progesterone and progesterone [50]. In the placenta, hCG is produced by the extravillous trophoblast and the villous trophoblast (Figure 2). Its production is closely related to the development and turnover of the trophoblast, and mainly the ST (Figure 1B). It modulates the differentiation of the VCT into the ST, and therefore its production, in an autocrine loop [41,42,43]. hCG binds to a specific receptor (LHCGR) that also recognizes LH. The binding of hCG to LHCGR, which is coupled with G proteins, modulates steroidogenesis mainly via the cAMP/PKA but also via the inositol phosphate pathways [23,50,51,52]. LHCGR is present in the placenta in truncated form in the VCT and in long form in the ST. During pregnancy, only the dimeric form of hCG has significant biological activity, which is reduced in its truncated forms in hCGn [44]. The main function of hCG is to promote progesterone synthesis by the corpus luteum, maintaining the quiescence of the myometrium and pregnancy until the placenta itself ensures progesterone production. High levels of hCG also interact with the TSH receptor, enhancing maternal thyroid function [53,54]. At the maternal–placental interface, hyperglycosylated forms of hCG and free hCGβ exert paracrine actions by promoting invasion and angiogenesis in the uterine wall [55,56,57] and maternal immune tolerance [48,58,59]. 

hCG anomalies in the context of fetal trisomy 21 are both qualitative and quantitative. In trisomy 21, maternal hCG and free hCGβ levels rise concomitantly, with an increase in their truncated and hyperglycosylated forms [60,61]. The presence of an extra chromosome 21 cannot directly account for the over-expression of the genes coding for hCG because it is not located on chromosome 21. No extraplacental production of hCG has been demonstrated, but an abnormal modulation of transcription cannot be excluded [62,63]. In trophoblastic cell cultures, an increase in hCGβ transcripts and, to a lesser extent, in hCGα transcripts has been observed, and results either from transcriptional activation or from an increase in transcript stabilization. Serum hCG levels in trisomy 21-affected pregnancies are significantly higher than in normal pregnancies [7] and the chorionic expression of ß CG mRNA directly correlates with high serum hCG levels [64]. Chromosome 21 may bear one gene encoding for a transcription factor that induces the synthesis of hCGβ [65]. ß CG promoters may differ between trisomy 21-affected pregnancies and normal pregnancies. The sequence and amino acid composition of intact hCG remain unchanged [66], while levels of urinary β core are increased in trisomy 21-affected pregnancies, reflecting an increase in both hCG levels and catabolism [67]. 

Several studies have highlighted the hyperglycosylation of hCG in trisomy 21. Cole et al. described the presence of a hyperglycosylated hCG (H-hCG) which they named invasive trophoblast antigen (ITA), as it is also produced by undifferentiated invasive cytotrophoblasts in choriocarcinoma [25,68]. This H-hCG displays a greater number of tri-antennary N glycan chains and its O glycan chains are also larger. It may be secreted physiologically during normal pregnancy (particularly in the first trimester), but its levels are low and fall below 3% of total hCG thereafter. It has been suggested that the major source of this H-hCG in maternal blood during the first trimester could be endovascular EVCT [33] (Figure 2). H-hCG levels are often elevated in maternal serum in trisomy 21-affected pregnancies [56,61,68]. Trisomy 21 is likely to be associated with the abnormal glycosylation of hCG. Indeed, we have shown that the in vitro differentiation of VCT isolated from trisomic 21 placentae is defective and associated with abnormal glycosylation hCG features, leading to a highly acidic and less bioactive molecule [69,70]. In vivo analysis of maternal hCG circulating glycoforms confirms this abnormal glycosylation [71]. At least two enzymes involved in the glycosylation pathway may be affected: sialyl-transferase-1 and fucosyl-transferase-1. In trisomy 21-affected pregnancies, the synthesis and the expression of the mature receptor of hCG, LHCGR, are modified [64,69] and a soluble fraction of LHCGR may circulate in the maternal blood [72]. Taken together, these changes may lead to abnormal hCG signaling and the clearance of hCG, resulting in high maternal serum levels in trisomy 21. 

#### 2.1.2. Pregnancy-Associated Plasmatic Protein A

PAPP-A is not specific to pregnancy as it is also detectable in men and non-pregnant women, but at 100 to 1000-fold lower levels [73,74]. In maternal blood, 99% of PAPP-A circulates at very high levels as a heterotetrameric covalent complex (ht PAPP-A) with the precursor of eosinophil major basic protein (proMBP) [75,76] (Figure 3). 

The human trophoblast synthesizes PAPP-A and its inhibitors, the precursor of eosinophil major basic protein (proMBP) and the stanniocalcin (STC): PAPP-A circulates in the maternal blood as an inactive complex with the proMBP. The dimeric PAPP-A is a protease expressed on the trophoblast membrane and involved in the invasion and modulation of insulin-like growth factor (IGF) activity as it cleaves its binding protein (BP).

PAPP-A is a homodimeric glycoprotein of 1547 amino acids with 14 N and 7 O glycosylation sites [77]. It is secreted as a precursor encoded by a single gene located on chromosome 9. Besides hormonal proteins, several other proteins are secreted by the placenta and referred to as pregnancy-associated plasma proteins (PAPP) types A, B, C, D, or E [78], the most widely studied being type A (PAPP-A) also known as pappalysin-1. It may be detected in its monomeric or dimeric (d PAPP-A) forms in both reproductive tissues (ovaries, endometrium, etc.) and in non-reproductive tissues and organs such as kidneys [74,79,80]. During pregnancy, it forms an equimolar heterotetrameric covalent complex (2:2) of 500–700 kDa with a small glycoprotein of 222 amino acids, proMBP (38 kDa), encoded by a single gene located on chromosome 11 [75,76]. The glycan cup of proMBP is much larger than that of PAPP-A (38%), consisting essentially of O glycan. As proMBP can bind different molecules and is synthesized in excess, it also circulates in maternal serum linked to angiotensinogen or C3dg fraction of the complement [81]. During pregnancy, genes encoding for PAPP-A and proMBP are strongly expressed in the placenta. Transcripts can be detected in the trophoblast but also in the decidua, with mRNA content of PAPP-A and proMBP increasing until term [82,83,84,85]. PAPP-A is expressed by the EVCT and VCT, while proMBP tends to be preferentially expressed by the EVCT and decidua [86,87]. dPAPP-A is weakly expressed in the ST at term, and serum PAPP-A levels start to rise from 4 WG until the end of pregnancy in line with increasing ST mass and placental volume [88,89,90]. Serum concentrations of proMBP display a similar profile, with a significant increase during the first trimester [91]. Glycosylation of these molecules also varies during pregnancy, with a decrease in sialylation [92].

Neither the cellular functions of PAPP-A and proMBP nor their regulation are yet fully understood in the placenta [93,94]. PAPP-A is a metalloprotease belonging to the metzincin superfamily of zinc peptidases. It cleaves binding proteins of the family of insulin-like growth factors, mainly IGFBP-4 but also IGFBP-2 and IGFBP-5, in many tissues [74,93,94,95,96], modulating the bioavailability of IGF-I and IGF-II (Figure 3). Via the activity of IGFs, PAPP-A modulates both cell proliferation and differentiation, and, via its proteasic activity, it modulates invasion [97]. PAPP-A has endocrine effects on glucose metabolism and maternal adaptation to pregnancy via IGFs [98]. PAPP-A is only active in its dimeric form, dPAPP-A, which is expressed in the cytotrophoblast membrane. It is an enzyme that forms a complex with cell-surface heparin sulfate proteoglycans to interact with cells and the extracellular matrix. PAPP-A activity is inhibited by the binding to proMBP and stanniocalcin [99,100,101,102]. Its catabolism and its physiological functions during pregnancy remain a matter of debate. PAPP-A production and activity are likely to be modulated by AMPc, cytokines, growth factors, hormones, and its endogenous inhibitor stanniocalcin, in different tissues in vitro, in association with tissue injury and inflammation [74,80,93]. Progesterone may positively modulate PAPP-A production, promoting the proliferation of the trophoblast, whereas PPARγ inhibits PAPP-A expression [103,104].

In the context of fetal trisomy 21, maternal serum levels of PAPP-A are lower during the first trimester of pregnancy [105]. However, no significant decrease in placental PAPP-A mRNA levels has been noted in the context of trisomy 21 [106]. It is also observed in other types of aneuploidy (trisomies 13 and 18 in association with a small placental mass and in the Cornelia Delange syndrome) [107,108]. This decrease is temporary, as it disappears after the first trimester of pregnancy when the ST becomes the major source of PAPP-A production and when the placenta contributes markedly to proMBP levels (Figure 2) [88,91]. A sharp reduction in PAPP-A maternal serum levels during pregnancy is often associated with abortion or premature birth, reflecting defective trophoblastic invasion and placentation [109,110]. 

#### 2.1.3. Inhibin A

The placenta, i.e., trophoblastic cells, mesenchymal cells, and endothelial cells, produces different amounts and types of growth factors [111,112]. These factors mainly act in an autocrine or paracrine manner, as they are involved in placental development [113]. A few of these markers have been measured in maternal blood for the screening of trisomy 21 angiogenic factors, such as placental growth factor (PlGF), which was more recently described as being decreased in trisomy 21-affected pregnancies [114,115]. However, the main growth factor measured in maternal screening for trisomy 21 is inhibin A, which belongs to the transforming growth factor beta (TGFβ) superfamily as other inhibins (B, C, D, and E), activins, and follistatin [116,117]. Inhibins are heterodimeric glycoproteins, made up of a common α-subunit and a specific β-subunit (five isoforms βA, βB, βC, βD, and βE), which inhibit activin activity [118]. Activins (A, B, and AB) are homodimers or heterodimers composed of two inhibin β-subunits (βA or βB), which can be associated with a carrier, the monomeric glycoprotein follistatin. All these proteins constitute a heterogeneous pool of circulating dimeric glycoproteins ranging from 30 to 110 kDa. Human α-subunit results from the cleavage of a precursor, which is encoded by a single gene located on chromosome 2 [119,120]. It contains three N glycans and its production is up-regulated by cAMP. β subunits also result from proteolysis of precursors, after cleavage of the β–C terminal element of 13 kDA. βA and βB subunits are encoded by two genes located on chromosomes 7 and 2, respectively, and expressed in the placenta. The βA subunit has one N glycan, whereas the βB subunit is not glycosylated. Cysteine residues are crucial to the formation of the three-dimensional protein structure, which is characteristic of all the members of the TGFβ superfamily, and which is responsible for their biological activity. In vitro studies have suggested that, as for hCG, glycosylation is an important driver for both the assembly and function of inhibins and activins [121]. Inhibin A inhibits the action of activin A on its receptors, whereas follistatin regulates the bioavailability of activin A [122,123]. Inhibin A antagonizes the action of the mature dimeric activin A that acts mainly via the type II (ActRII) serine–threonine kinase receptor, inducing Smad signaling. The binding of inhibin A to ActRII is facilitated by a membrane proteoglycan, the betaglycan. In the placenta, ActRII and betaglycan are expressed in the ST and the endothelial cells in both the villi and the decidua [124]. Therefore, inhibin A in the placenta acts in an autocrine and paracrine way, modulating the actions of activin A on decidua cells, vascular endothelial cells, and trophoblast. The inhibin/activin/follistatin system is involved in the functional differentiation of the trophoblast in relation to estradiol and hCG [125]. Inhibin A is a potent antagonist of activin-mediated steroidogenesis and hCG production by the ST [126]. It may also be involved to a minor extent in triggering labor [124]. Activin A increases matrix metalloproteinase (MMP) 2 expression and cell invasion in the human trophoblast, whereas inhibin A blocks this process [127].

Inhibins, activins, and follistatin are expressed in different tissues, i.e., brain, and breast, but mainly in reproductive tissues, i.e., testes, ovary, and placenta [118,123]. In a full-term placenta, activin A and inhibin A are found at high levels in the ST, as well as follistatin in smaller amounts. Indeed, the high inhibin A levels produced by the placenta increase during the second trimester to reach a peak in the third trimester [128,129,130,131]. Inhibin A seems to be produced by the ST rather than by the CT and its synthesis depends on the α subunit rather than on the βA subunit mRNA. Inhibin and activin production is regulated by numerous factors that act locally on trophoblast differentiation such as hCG, EGF, CRH, oxygen, AMPc, activin A itself, and cytokines.

Maternal serum inhibin A and activin A have been studied in normal and pathological pregnancies [132]. In normal pregnancy, inhibin A levels increase mainly during the third trimester, whereas inhibin B levels are hardly detectable [131]. Abnormal levels of inhibin A have been described in pathological pregnancies, with low levels being associated with miscarriage or trisomy 21 [11] and high levels being associated with preeclampsia [132,133]. Only inhibin A is used as a trisomy 21 serum marker during the second trimester, mainly because its measurement is simpler and more specific than those of activin A and follistatin. Van Lith was one of the first to report an association between elevated maternal serum inhibin A and trisomy 21-affected pregnancies in 1992 [116]. In cases of trisomy 21, inhibin A levels increase by around two-fold as hCG [11,134,135]. This increase seems to result from an increased trophoblastic synthesis since an up-regulation of inhibin α-subunit mRNA levels in placenta from trisomy 21-affected pregnancies has been demonstrated and associated with a higher placental concentration of inhibin A at the protein level [134]. Molecular forms of inhibin in trisomy 21 did not differ from normal pregnancies in the maternal serum in contrast with the placenta [136]. Interestingly, Kipp et al. compared potential regulators of the α-inhibin promoter between unaffected and trisomy 21-affected full-term placental tissues [137]. Their findings highlighted a reduced expression of WT1, an inhibin α-subunit repressor, which may explain the higher placental production and the elevated maternal serum levels of inhibin A. We showed in vitro that T21 villous mesenchymal cells secrete less activin A than non-affected mesenchymal cells [138]. Moreover, the use of recombinant activin A stimulates trisomic 21 trophoblast fusion, whereas follistatin and blocking activin A antibodies inhibit this fusion. These results suggest a putative role of activin A in ST formation and functional defects observed in trisomic 21 placentas.

Other markers measured in maternal serum come from the fetus, but some steps in their production, secretion or release into the maternal circulation require the cooperation of the placenta, and mainly the ST.

### 2.2. Markers of Fetal Origin: The Alpha-Fetoprotein

Alpha-fetoprotein (AFP) is a glycoprotein that was discovered in 1956 by Bergstrand and belongs to the oncofetal proteins group [139,140]. It is one of the four members of the albumin superfamily, sharing a high degree of homology, as they derive from the same ancestral gene [141,142]. 

This glycoprotein (MW 67–74 kDa) is composed of a single chain of three domains and 590 amino acids with 15 disulfide bridges, which migrate in the alpha1 band [143]. It is encoded by a single gene located on the long arm of chromosome 4 [144]. It contains no more than 5% of glycans with several putative N and O glycosylation and phosphorylation sites, depending on the producing tissue [145]. During pregnancy, it is initially synthesized by the yolk sac until 13 WG and then by the fetal liver until the end of pregnancy (Figure 4) [146]. 

A small proportion of AFP is produced by both the digestive tract and the kidneys during the very early stages of pregnancy, and a transient production by the trophoblast has also been suggested [147,148]. AFP is the first alpha-globulin to appear in fetal blood during ontogenesis. Its level rises to reach a peak of 3 to 4 mg/mL at around 13–15 WG and then declines as fetal hepatic synthesis switches to albumin [149]. After delivery, its level in neonatal blood falls drastically during the first month of life to the undetectable level observed in children and adults [150]. In early pregnancy, AFP is likely to pass from the fetal circulation through the unkeratinized fetal skin into the amniotic fluid. Thereafter, fetal serum AFP is excreted by the immature fetal kidneys via the urine in amniotic fluid, peaking at around 14 WG at much lower levels (μg/mL) [151]. Then, renal AFP filtration decreases with kidney maturation and therefore its amniotic levels fall until the end of pregnancy. AFP reaches the maternal circulation through the membranes and placenta, explaining the high concentrations found in the intervillous space [152]. This transfer is very low, as maternal serum AFP levels are several hundred-fold lower (ng/mL) than those observed in amniotic fluid, reaching a plateau at about 32 WG. The mechanisms involved in this placental transfer are still unclear and may involve lectins [152,153,154]. The regulation of AFP production is also still poorly understood [140,155,156]. AFP is associated with the cell cycle and is synthesized during both the G and S phases of cell division [157]. Several transcription factors appear to be involved, such as the hepatocyte nuclear factor 1 (HNF1), nuclear factor 1(NF1), and the C enhancer binding protein(C/EBP). Some of them are specifically detected in the fetal liver such as Nkx2,8 and the fetal transcription factor (FTF) [158]. Because the HNF1/NF1 ratio varies during pregnancy, this may explain the decrease in fetal AFP synthesis. Activation of the “silencer” located between the enhancer and AFP promoter will also lead to an increase in albumin synthesis while AFP synthesis decreases [159]. The biological functions of AFP are still debated, but it is likely to play an important role at the fetal–maternal interface [140,156,160,161,162]. AFP is involved in many physiological processes, such as growth control, cellular proliferation [163], inflammation, immune modulation [164,165], and apoptosis [166]. Like albumin, its binding capacities allow it to modulate the bioavailability of endogenous ligands necessary for embryofetal development (e.g., fatty acids in neural tissue) or exogenous potentially toxic ligands (e.g., bilirubin, estrogens, heavy metals, retinoids) [167,168,169].

AFP levels in amniotic fluid and maternal serum can be abnormally increased in many fetal defects, making AFP and some of its glycosylated isoforms a sensitive but not specific marker of neural tube defects [162,170,171,172]. In trisomy 21, maternal serum AFP levels fall to about 0.8 MoM, and can therefore be used as a T21 biomarker during the second trimester [9,173].

However, several reports have suggested that this decrease in maternal serum levels of AFP could be detected during the first trimester of pregnancy [174]. A delay in renal maturation and AFP excretion into amniotic fluid has been discussed in trisomic 21 fetuses [175]. Decreased AFP production by the liver of trisomic 21 fetuses has been suspected by the study of AFP transcripts but has not been confirmed at the protein level [176,177]. AFP glycosylation is modified in trisomy 21 [178,179,180]. This modification may decrease its transfer across the membranes, leading to lower maternal serum levels in trisomy 21. The association of normal fetal hepatic AFP levels with increased placental AFP levels in trisomy 21 is in favor of reduced placental transfer [181]. It may also result from impaired villous vascularization or maturation in the context of aneuploidy.

### 2.3. Markers of Feto–Placental Origin: The Unconjugated Estriol

Finally, estriol (E3), a steroid hormone secreted by the placenta, which belongs to the estrogen hormones, such as estradiol (E2) and estrone (E1), can also be used as a maternal serum marker [181,182]. Unlike other tissues producing estrogens, the human placenta lacks cytochrome P450 17alpha-hydroxylase-17:20 lyase, which is required to convert progestins into androgens, the precursors of estrogens (Figure 5) [183,184,185,186,187]. Thus, the placenta participates in the enzymatic conversion of fetal 16α-OH-DHEA-S into E3, by sulphatase (STS), and aromatase activities [188,189]. Estrogen production is regulated by several hormones, i.e., hCG, calcitriol, insulin leptin, and cAMP, which are also involved in cytotrophoblast differentiation [189,190,191]. Estrogens diffuse into both the fetal and maternal compartments. E3 is detectable in maternal serum as early as 8 WG and its levels increase until the end of gestation [192,193]. Aside from pregnancy, E3 is produced at very low levels as a hepatic catabolite of estradiol. E3 mainly circulates in conjugated forms after the action of the maternal liver. Only 10% of circulating E3 is unconjugated (uE3) and likely to bind to the sex hormone-binding globulin (SHBG) [194]. The half-life of the unconjugated form is 20 min, with maternal renal and/or biliary excretion.

E3 functions are assimilated to those of E2 during pregnancy and act in an endocrine, paracrine, and autocrine manner involving estrogen nuclear receptors (ER): ERα (66 kDa) and ERβ (53 kDa) [195]. Cytotrophoblasts mainly express ERα while the ST mainly expresses ERβ. E3 interacts mainly with ERβ [196]. It is considered to be a weak estrogen for all the biological effects of E2, i.e., endometrial maturation, maternal metabolic adaptation to pregnancy inducing hepatic functions, pituitary growth, and proliferation of mammary epithelium in preparation for breastfeeding. E3 appears to be specifically involved in the angiogenesis and regulation of placental vascularization, designed to increase uteroplacental blood flow, trophoblast turnover, and therefore feto–placental exchanges [195,196,197,198]. Thus, during the third trimester of pregnancy, E3 is positively correlated with neonatal weight and size at birth and placental weight [199].

uE3 is the only steroid hormone that may be used in clinical practice for the maternal screening of T21, but its analytical determination is difficult, therefore restricting its use [17,200,201,202]. In trisomy 21, levels of uE3 are significantly reduced in maternal serum in the second trimester of gestation [10,202]. This decrease is also observed in the placenta, maternal urine, and amniotic fluid, since the end of the first trimester [203,204]. It correlates with a decrease in maternal serum levels and results from a lower production by the placenta. This seems not to be due to a decrease in the placental STS activity and aromatase activity but this has not been fully studied in trisomy 21. Interestingly, DHEA-S, an estrogens precursor specifically produced by the adrenal gland, is lowered in maternal serum, placental tissue, and fetal liver in trisomy 21, suggesting a potential adrenal dysfunction [205]. This hypothesis is supported by the lower DHEA levels observed in the serum of trisomic 21 adults [206].

## 3. Cellular Markers and Placenta

Noninvasive prenatal screening (NIPS) is indicated for trisomies 21, 18, and 13 screening, especially for T21. It is based on cell-free DNA (cfDNA) analysis in the maternal blood, but many different cells and feto–placental cellular components have been characterized in the maternal compartment [207,208].

### 3.1. Cellular Markers and Cell-Free Fetal DNA

The cross-placental trafficking of fetal cells and the release of placental fragments into maternal blood is a frequent phenomenon occurring throughout pregnancy. Different fetal cells may be detected, and the presence of fetal erythrocytes has been characterized in maternal serum [209,210]. They are likely to be released from endothelial capillaries inside placental villi either by accidental breakage of the villi or by vessel breaching. Schmorl was one of the first to describe the presence of placental trophoblastic cells in the lungs of women who had died from preeclampsia [211]. Villous trophoblast tissue fragments are physiologically deported into the intervillous space and thereafter into the maternal circulation by the shedding of syncytial knots [212]. This results from the normal apoptosis of the ST and turnover of the placental villi [213,214,215]. Trophoblast tissue fragments are increased in complicated pregnancies, particularly in preeclampsia, reflecting ST necrosis. EVCT, which invades the uterine wall, reaching the spiral arteries, may also contribute to endovascular trophoblast [216]. Placental components such as exosomes, cell-free DNA, and RNA, have been identified in maternal blood [217]. Cell-free feto–placental DNA is detectable at increasing levels in the maternal circulation since 5 WG but represents less than 20% of the total cell-free DNA circulating in maternal blood [218]. It consists of an altered DNA, mainly of placental origin, which disappears very quickly after delivery in contrast to fetal cells, which can survive several weeks or years post partum. Feto–placental DNA circulates mainly in cell-free form, whereas free feto–placental RNA is mostly encapsulated in large, sealed membrane particles. It mainly results from the placental physiological turnover with the release of syncytial knots in the maternal circulation, independently of placental size [213,219]. However, accidental breakage or necrosis, especially in complicated pregnancies, may contribute to their release. There is an apoptotic release from ST associated with cell-free DNA release but also with the release of exosomes and small membrane fragments (STBM) [214] An increased apoptotic process and syncytial necrosis in trisomy 21 has been suggested to explain the global increase of placental cellular components, such as cell-free feto–placental DNA, in maternal serum.

Increased levels of fetal cells and fetal DNA have been reported in aneuploidies, especially in trisomy 21, by the analysis of enriched fetal cells and PCR [215,218,219,220,221,222]. Moreover, seven miRNAs have been shown to be up-regulated in trisomic 21 placentas, three of them (miR-99a, miR-1125b, let-7c) being located on chromosome 21. They are likely to modulate different cell biological processes and genes involved in placental development (e.g., LEP, INHA) [223].

### 3.2. Placenta and Trophoblastic Villi in Trisomy 21

There is a defective placental development in trisomy 21 affecting trophoblast proliferation, invasion, and differentiation, vasculogenesis, and mesenchymal development (Figure 4). All these defects are likely to influence endocrine, transfer, and exchange placenta functions. Placentae in trisomy 21-affected pregnancies exhibit immature or dysmature villi (hypotrophic or hydropic) with mesenchymal dysplasia, stromal edema, hypovascularization, trophoblastic hypoplasia, intrastromal trophoblastic cysts, calcification, and fibrin deposits [224,225,226,227]. The persistence of a two-layered VCT is also observed, suggesting a delay in villous maturation. However, these lesions do not always affect the whole placenta and may evolve during pregnancy. In addition, they are not specific for trisomy 21, as they may also be observed in other types of aneuploidy [228]. This might explain why maternal serum markers are not always abnormal in trisomy 21, depending on the degree of placental injury, and why these markers can be abnormal in trisomy 18 and trisomy 13.

The formation of the ST is defective in trisomy 21. Gap junctions between VCT are likely to be affected by miRNAs that are up-regulated in trisomy 21. Thus, VCT aggregates but poorly fuses to form an ST expressing low levels of LHCGR [229,230,231]. This defective differentiation is associated with the production of an abnormally glycosylated hCG with decreased steroidogenic activity [69]. This weakly bioactive hCG and abnormal LHCGR expression could be involved in defective ST formation [64,70]. Fibroblasts isolated from the mesenchymal axis of trisomic 21 villi display low levels of activin A and increased levels of follistatin, participating in the decrease in VCT fusion. In addition, trophoblasts in trisomy 21 show an abnormal oxidative status, impairing both VCT fusion and differentiation. This status is determined by the balance between ROS production and their degradation by antioxidant enzymes such as superoxide dismutases (SODs). The Cu, Zn-SOD (SOD-1) is encoded by a gene that is located on chromosome 21. In trisomy 21, there is indeed an increased expression and activity of SOD-1, resulting in abnormal oxidative status, and impairing ST formation [229,232]. This abnormal oxidative status can lead to increased apoptosis and necrosis of the ST [232,233].

## 4. Conclusions

Trisomy 21 is the main genetic cause of mental retardation. This aneuploidy is compatible with post-natal survival and associated with a wide spectrum of phenotypic features, some of them being detectable by fetal ultrasound scanning and being part of prenatal screening [1,3]. NIPS based on cfDNA analysis is now widely used as it has high performance (PPV ≥ 90% and NPV ≥ 99%) to screen for trisomy 21. It is also informative to screen for trisomy 18 and trisomy 13 even with lower performance compared to 21, leading to a global decrease in invasive procedures for karyotyping [16,207,234]. NIPS is likely to spread to other fetal chromosomopathies and monogenic diseases, and recent guidelines recommend offering NIPS to all pregnancies [235,236]. However, due to its current cost, NIPS is still used in many countries as a second-tier test following maternal serum screening methods.

The serum profiles observed in the case of fetal trisomy 21 are characterized at the end of the first trimester by decreased PAPP-A and increased free hCGβ levels; in the second trimester, it is characterized by a decrease in the AFP and uE3 levels and an increase in the hCG, free hCGβ and inhA levels [15,172] (Figure 6). These abnormal profiles are likely to be observed all through pregnancy even if not always discriminant at other stages [174,203,229]. This suggests that the molecular mechanisms underlying these abnormal serum levels are present all through pregnancy. In trisomy 21-affected pregnancies, the placenta is likely to be both immature and affected by increased oxidative stress and apoptosis [233]. However, maternal serum markers are mainly but neither consistently disrupted (in the case of fetal trisomy 21) nor specifically, as a decreased PAPP-A can be observed in trisomy 18 and an increased hCG can be observed in preeclampsia [110,237]. In addition, other placental markers have been described as abnormal in the case of fetal aneuploidy and have also been proposed in the screening of preeclampsia such as placental growth factor (PlGF) [238,239]. The molecular mechanisms underlying these abnormal serum levels are likely nonspecific to any of those diseases. Maternal serum biomarkers both reflect the alteration of the trophoblast and point out the key role of the placenta in prenatal screening.

The presence of an extra chromosome 21 has a potential impact on the overall genomic expression of the fetus and the placenta [240,241]. This could be mediated by exosomes and miRNAs that are abnormally expressed in trisomy 21 and involved in the regulation of transcription, glycosylation, and apoptosis in the trophoblast [242,243]. The trophoblast can thus be variably dysregulated in its differentiation and functions, explaining both the high frequency of early miscarriage in pregnancies affected by trisomy 21 and the abnormal marker pattern in maternal serum. Many diseases affecting trisomic 21 adults may result from biological disorders such as abnormal glycosylation, oxidative stress, and mitochondrial dysfunction beginning in fetal life [244,245,246].

## Figures and Tables

**Figure 1 ijms-24-07669-f001:**
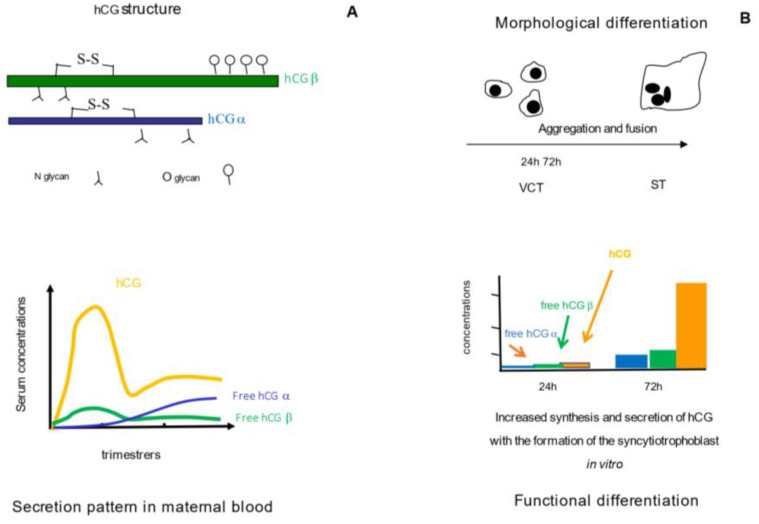
Structure and secretion of hCG in vivo (**A**) and in vitro (**B**). hCG is a dimeric glycoprotein composed of a common α subunit (hCGα), and a specific ß subunit (hCGβ). The secretion of hCG and its free ß subunit in maternal blood increase during the first trimester of gestation, and decrease thereafter; the secretion of its free α subunit increases all through pregnancy in line with placental mass (**A**). In vitro and vivo, the morphological differentiation of the villous cytotrophoblasts (VCT) into a syncytiotrophoblast (ST) is mainly associated with functional differentiation, as assessed by the increasing production of hCG and its subunits (**B**).

**Figure 2 ijms-24-07669-f002:**
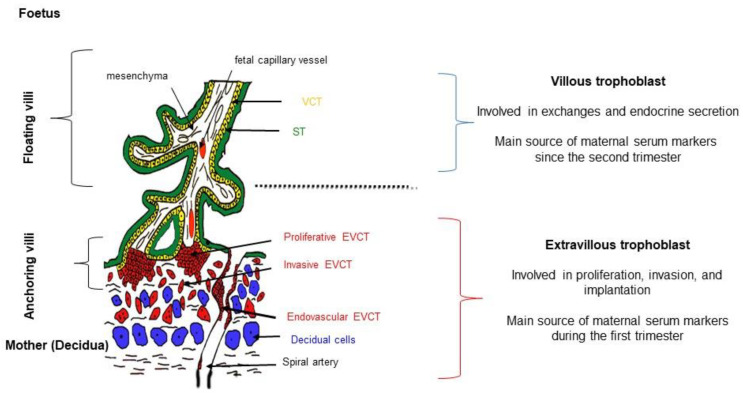
Human placental definitive villi and trophoblastic tissue. From the end of the first trimester of gestation, the human placenta is composed of two types of villi: the anchoring villi with extravillous cytotrophoblasts (EVCT) that proliferate, migrate, and invade the maternal uterine wall to reach the spiral artery; the floating villi composed of villous cytotrophoblasts (VCT), aggregating and fusing to form the syncytiotrophoblast (ST) on the borders of floating villi in contact with maternal blood as from 10 to 12 WG. EVCT is likely to be the major source of maternal serum markers in early pregnancy, whereas ST becomes the major source at the end of the first trimester of gestation.

**Figure 3 ijms-24-07669-f003:**
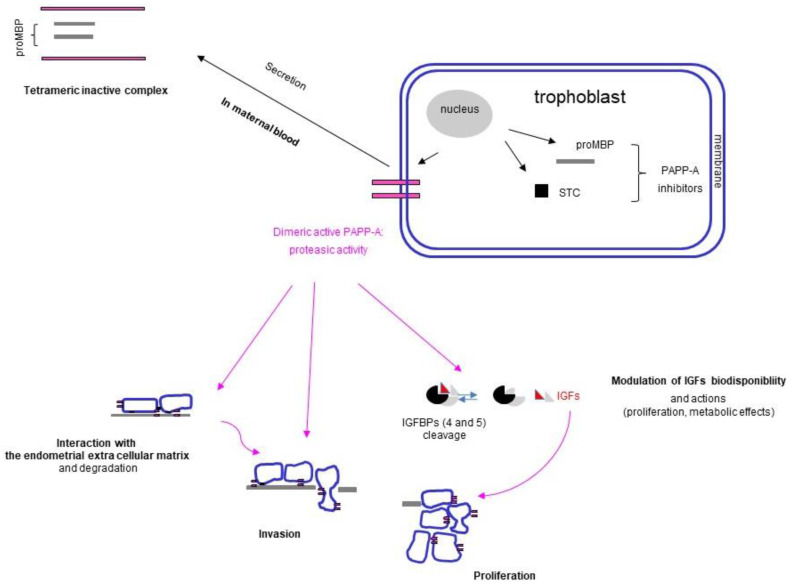
Structure and main functions of pregnancy-associated plasmatic protein A (PAPP-A).

**Figure 4 ijms-24-07669-f004:**
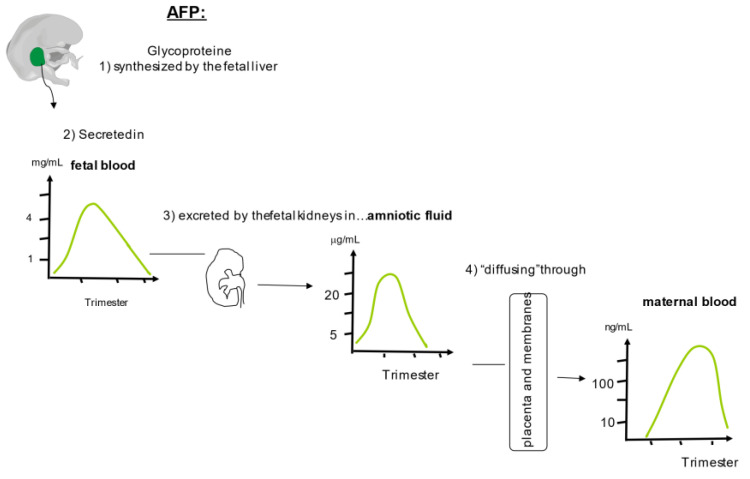
Synthesis and secretion of alpha feto protein (AFP) during pregnancy. AFP is mainly synthetized by the fetal liver and secreted in fetal blood. It is then excreted by the immature fetal kidneys via urine in amniotic fluid, peaking at around 14 WG. Thereafter, renal AFP filtration decreases with kidney maturation and therefore its amniotic levels fall until the end of pregnancy. AFP reaches the maternal circulation through the membranes and placenta. This transfer, which may involve lectins, is very low and reaches a plateau at about 32 WG.

**Figure 5 ijms-24-07669-f005:**
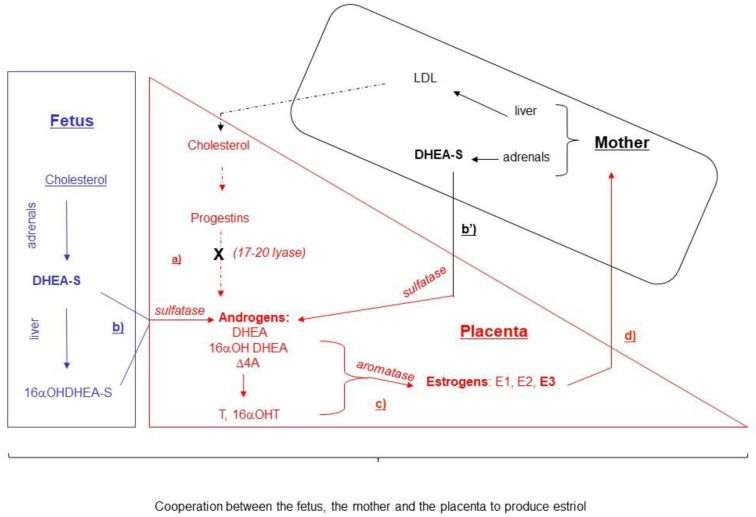
Synthesis and secretion of estriol during pregnancy. The human placenta lacks cytochrome P450 17alpha-hydroxylase-17:20 lyase, which is required to convert progestins into androgens (a). Thus, the placenta participates in the enzymatic conversion of fetal 16α-OH-DHEA-S, DHEA-S (b), and the maternal DHEA-S (b’) into E3, by sulfatase (STS), and aromatase activities (c). Estrogens diffuse into both the fetal and maternal compartments (d). estriol (E3), estradiol (E2), estrone (E1), dehydroepiandrostenedione (DHEA), sulfate of dehydroepiandrostenedione (SDHEA), delta 4 androstenedione (Δ4A).

**Figure 6 ijms-24-07669-f006:**
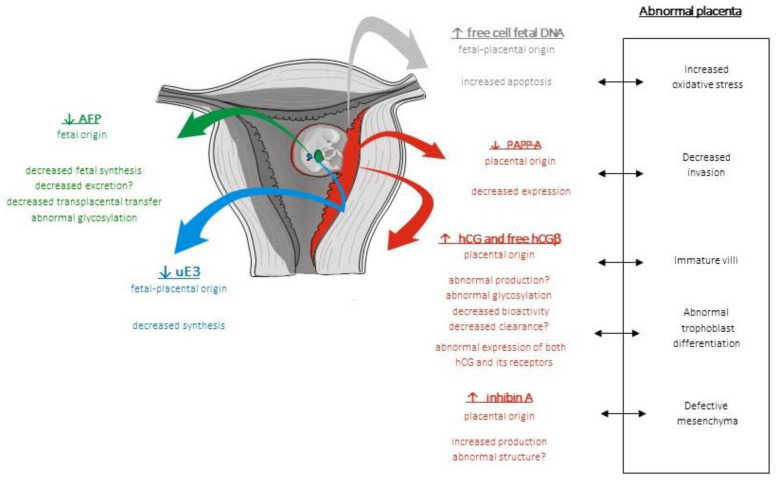
Physiopathology of maternal serum markers in trisomy 21-affected pregnancies. The human placenta releases soluble markers into the maternal blood as well as cells and cell-free DNA. Soluble markers can be of fetal origin, such as the alpha feto protein (AFP), of feto–placental origin, such as the unconjugated estriol (uE3), or of placental origin, such as the human chorionic gonadotropin (hCG) and its free beta subunit (hCGβ), the pregnancy-associated plasmatic protein A (PAPP-A), and the inhibin A (inhibin A). In trisomy 21-affected pregnancies, both placental and fetal functions are disrupted, leading to the decrease (↓) or the increase (↑) of their concentrations in maternal blood.

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
