# Peer review of "Biochemical Screening for Fetal Trisomy 21: Pathophysiology of Maternal Serum Markers and Involvement of the Placenta"

_ijms, 2023, doi:10.3390/ijms24087669_

Round 1

Reviewer 1 Report

The review article titled "Biochemical screening for fetal trisomy 21: pathophysiology of 2 maternal serum markers and involvement of the placenta" is very interesting and very educative for PhD students, medical specialists, and researchers. Some parts of the review are difficult to understand, especially Part I-1-1 Human Chorionic Gonadotropin. Therefore, the authors might consider restructuring this part to make it more fluid for readers. My suggestion is to simplify the biochemical structure and regulatory hypothesis. Line 84 - Figure 3A in parentheses is not needed.

The figures are informative except for figure #1. i suggest deleting it or creating a new figure that is more informative and better designed. All figures should be better designed and described (Figure 5). Figure 4 could be the last or first one, because this figure summarizes the physiopathology of markers in trisomy 21.

 Reference 242 is incomplete.

Author Response

We thank the reviewer 1 for his comments. We have integrated (in red) all the remarks and suggestions in the revised version of the manuscript.

-English language and style have been checked

-Part I-1-1 Human Chorionic Gonadotropin has been restructured to make it more reader friendly.

-The description of biochemical structures and hypothesis on regulation mechanisms has been shortened.

-Line 84 –Figure 3A: brackets have been deleted

- Figure 1 has been deleted

-All figures and their caption have been improved

-Figure 4 has moved to Figure 6 and is now the last figure of the paper.

Reviewer 2 Report

Dear authors,

I read with great interest the manuscript, which falls within the aim of this Journal. In my honest opinion, the topic is interesting enough to attract the readers’ attention. Nevertheless, authors should clarify some points and improve the discussion, as suggested below. Authors should consider the following recommendations:

In my opinion you have to improve the paper refering in the text how the Cell free dna test itr really important to detecte aneupolidy and as maternal serum markers as well patients with high screening levels are at increased risk to develope preeclampsia so refer to0 the possible treatments as endoglin treatment.

Its could interesting also to refer to the possible increased risk in case of use of  contrast agents during pregnancy.

I suggest you to read and cite these articles:

Cell-Free Fetal DNA and Non-Invasive Prenatal Diagnosis of Chromosomopathies and Pediatric Monogenic Diseases: A Critical Appraisal and Medicolegal Remarks

The role of endoglin and its soluble form in pathogenesis of preeclampsia.

Contrast agents during pregnancy: pros and cons when really needed.

Author Response

We thank the reviewer 2 for his comments. We have tried to answer to all the raised issues in the revised version of the manuscript (in red).

-We have now clearly highlighted in the discussion the central role of cell free DNA for trisomy 21 but also other fetal aneupoloidies (trisomy 18 and trisomy 13) screening. We have also integrated the references suggested ( Cell-Free fetal DAN…, Gullo G et al., J Pers Med 2022).

-We have extended the interest of maternal serum markers and the involvement of placenta to preeclampsia in the discussion (new references 237-242).

-We have not integrated any  reference reporting an increased risk of trisomy 21 when using contrast agents during pregnancy. Indeed, the calculated risk and its value was not the topic of this article and there are many other maternal conditions more frequent than the use of  contrast agent (e.g. lupus , renal failure etc..) that may interfere with the biomarkers levels and the resulting calculated risk.